# Drying Kinetics of a Single Biomass Particle Using Fick's Second Law of Diffusion

**Jianjun Cai [1,2,*], Lingxia Zhu [2], Qiuxia Wei [2], Da Huang [1,3,*], Ming Luo [4,*] and Xingying Tang [5]**

1  Guangxi Key Laboratory of New Energy and Building Energy Saving, Guilin 541004, China
2  School of Architecture and Traffic, Guilin University of Electronic Technology, Guilin 541004, China
3  College of Civil and Architecture Engineering, Guilin University of Technology, Guilin 541004, China
4  School of Energy and Power Engineering, Jiangsu University, Zhenjiang 212013, China
5  Guangxi Key Laboratory on the Study of Coral Reefs in the South China Sea, School of Marine Sciences, Guangxi University, Nanning 530004, China
*  Correspondence: caijianjun@mails.guet.edu.cn (J.C.); dada_wong@glut.edu.cn (D.H.); mingluo@ujs.edu.cn (M.L.)

**Abstract:** Drying has been widely studied as a necessary process in biomass utilization. The steam diffusion law plays an important role in drying kinetics. The drying kinetics of a single biomass particle using Fick's second law of diffusion was studied in this paper. A parabolic relationship appeared between the critical moisture content and temperature. The critical moisture content decreased with the increase in drying temperature and the initial moisture content. The drying temperature had a significant effect on the effective diffusivity and coefficient of mass transfer during the dramatically falling period of the biomass drying process. However, it was affected by the effective diffusivity and coefficient of mass transfer during the slowly falling period. The initial moisture caused the opposite effect during the different periods. The normalized biomass moisture content generally increased with the increase in drying temperature, and decreased with the increase in initial moisture content. The initial moisture content had an effect on the normalized biomass moisture during the slowly rising period. Meanwhile, the drying temperature had an effect on the normalized biomass moisture during the whole period. The critical moisture content and the normalized biomass moisture content had negative relevant relationship. This study provides some valuable conclusions regarding the biomass drying process.

**Keywords:** drying kinetics; single biomass particle; initial moisture content; Fick's second law of diffusion

## 1. Introduction

As a renewable and environmentally friendly energy source, biomass (i.e., any organic non-fossil fuel) and its utilization has become increasingly important role worldwide [1]. Different thermo-chemical conversion processes, which include combustion, gasification, liquefaction, hydrogenation and pyrolysis, have been used to convert the biomass into various energy products [2]. Unfortunately, primary biomass often contains considerable amounts of water [3]. The presence of water has many negative effects on the performance of biomass and the development of conversion technology [4,5]. Therefore, the drying pretreatment of primary biomass is essential to improve the efficiency of biomass utilization. In recent years, most of the latest biomass utilization plants have been integrated with drying facilities [6,7]. Depending on the drying technology and the properties of the biomass, the drying time can take up to a month [8] using hot air or only a couple minutes with high temperature flue gas [9]. While biomass with a moisture level of 50~65 wt% can sustain combustion, the optimum moisture content is 8~15 wt% [10]. Brammer et al. showed the importance of drying biomass for small- to medium-scale biomass gasification plants for the production of heat and power, and verified that high levels of moisture content

within feedstock not only lowers the performance of the system, but also deteriorates the quality of the product gas [11].

Suherman et al. suggested that, in order to design an optimal drier, the drying kinetics of a single particle of the solid product must be known [12]. The dryer can be scaled up to a variety of types and dimensions from the single particle drying curves [13]. The drying kinetics mainly include the critical moisture content, the effective diffusivity, the coefficient of mass transfer, and the normalized drying curve [14–17]. Previous research has shown that the different types of dryness have different critical moisture contents, as well as normalized drying curves [15,17]. However, in terms of the existing literature, little attention has been paid to the influence of the drying temperature and the initial moisture content on single particle drying kinetics. In fact, the drying temperature and the initial moisture content play important roles in optimizing this technology [18,19]. Consequently, it is meaningful to explore the influence of the drying temperature and the initial moisture content on single particle drying kinetics.

Mathematical models of the drying process are generally divided into distributed models and lumped parameter models [20]. The distributed model considers both internal and external heat and mass transport, and can predict temperature and moisture gradients in the material. These models rely on Luikov equations [21], and the equations are derived from Fick's second law [22]. It uses the irreversible transport law of thermodynamics, and starts from the basic relationship of mass, momentum, and energy conservation, assuming that the temperature gradient and the concentration gradient work together for water diffusion. Finally, a set of partial differential equations are defined to describe the relationship between the internal heat and mass transfer in materials. In most drying processes, the influence of pressure is much less than the influence of the temperature and the moisture content on the drying process of materials, so the pressure equation is generally ignored. The lumped parameter model does not focus on the temperature gradient in the material. It assumes a uniform temperature distribution and that the material temperature is equal to the drying medium temperature. In the lumped parameter model, the uniform temperature distribution and the sample temperature coincide with the ambient temperature; these two assumptions can lead to some calculation errors [23]. These errors will be obvious at the beginning of drying, and reducing the thickness of the material will significantly reduce the calculation error. Therefore, this study used Fick's second law to build mathematical models of the drying process.

In this study, the drying kinetics of a single biomass particle using Fick's second law of diffusion were investigated using a hot air circulating oven. The drying events occurred in the drying process of the biomass. The kinetic parameters were obtained using drying curves, and the critical moisture content, the effective diffusivity, the coefficient of mass transfer, and the normalized drying curve were computed using those same kinetic parameters. This study aimed to gather useful data that could provide important references for the design and operation of a biomass dryer.

## 2. Materials and Methods

### 2.1. Materials

The biomass material used in the present study was a poplar biomass (wood chip), which was selected from a local furniture factory. The particles chosen for the drying experiments were 0.01 m × 0.01 m × 0.01 m in size. The pre-processing moisture content was decided by directly soaking the dried biomass particle in water. The final initial moisture content of biomass was determined by means of the AOAC method, no. 934.06 (AOAC, 1990). Biomass at high temperature (>200 °C) undergoes pyrolysis reactions with significantly morphological changes in color and volume. Whereas, in the drying temperature range (<150 °C), these changes are negligible. The proximate and ultimate analyses of the raw samples are shown in Table 1.

**Table 1.** Proximate and ultimate analyses of the raw samples.

| Proximate Analysis (wt %,DB) | | | Ultimate Analysis (wt %,DB) | | | | | | HHV |
|---|---|---|---|---|---|---|---|---|---|
| M | V | FC | A | C | H | O | N | S | MJ/kg, DB |
| 8.39 | 84.31 | 6.80 | 0.50 | 46.40 | 6.25 | 47.33 | 0.08 | 0.04 | 18.71 |

Note: DB, dry basis; FC, fixed carbon; HHV, higher heating value; M, moisture; V, volatile matter; A, ash.

### 2.2. Experimental Apparatus and Methods

Drying experiments were performed using a hot air circulating oven, as shown in Figure 1. The variable power range of the blower was 0.6 kW. The adjustable frequency range of the variable-frequency drive was 0–50 Hz. The variable power range of the electronic heated apparatus was 1–4 kW. The measuring range of the load cell was 0~2 kg, and its precision was 0.3%. The temperature and humidity of the hot air were maintained using hygro-thermometers (HD2101.1, DeltaOHM, Padova, Italy), which were equipped with temperature and humidity probes (HP474AC, Accuracy: humidity, $\pm$2.5%, temperature, $\pm$0.3 °C; air temperature and humidity measuring range: 0~100% RH and $-$40~150 °C). The speed of the hot air was determined using air speed transmitters (HD103T, Air speed Accuracy: $\pm$(0.04 m/s +2% of measurement); the air speed measuring range was 0.05~2 m/s by HD103T.

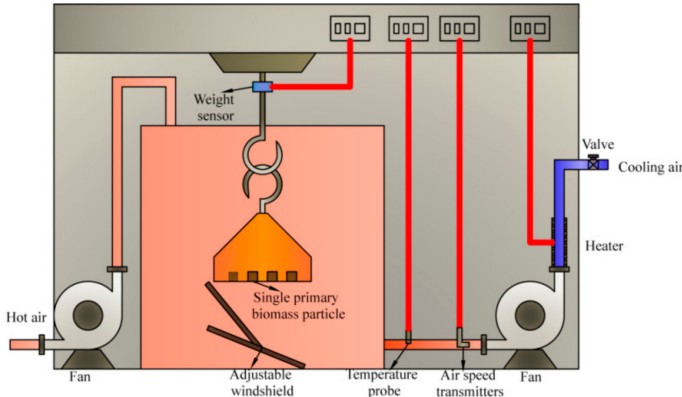

**Figure 1.** Hot air circulating oven.

Drying experiments were performed at seven temperatures (70, 80, 90, 100, 110, 120, and 130 °C). Considering the stability of weighing systems in drying experiments, and the speed of air (commonly, below 1.5 m/s) in industrial drying plants, the speed of the circulating air was set to 0.8 m/s. All experiments were replicated three times at each temperature and averages of weight loss were used.

### 2.3. Mathematical Modeling of Drying Curves

In order to quantify the performance of the drying rate, the moisture content (*M*) is defined as follows:

$$M_t = \frac{(m_t - m_\infty)}{m_\infty} \tag{1}$$

where $M_t$ is the moisture content at time *t*, kg $H_2O$/kg of dry matter; $m_t$ is the mass of the dried sample at time *t*, kg; $m_\infty$ is the mass of the completely dried sample, kg.

The moisture ratio (*MR*) is calculated using the following equation [24–26]:

$$MR = \frac{(M_t - M_e)}{(M_o - M_e)} \tag{2}$$

where $M_0$ is the initial moisture content (kg water/kg dry matter), and $M_e$ is the equilibrium moisture content (kg water/kg dry matter). The value of $M_e$ is equal to the moisture content

at the end of drying, at which the sample weight becomes constant with the drying time. When the temperature approaches 150 °C, the mass loss of the biomass is very small, and most water is removed. The value of $M_e$ is relatively smaller compared with $M_t$ and $M_0$, and hence, can be neglected. Therefore, the dimensionless moisture ratio, *MR*, can be simplified as follows [27,28]:

$$MR = \frac{M_t}{M_0} \tag{3}$$

The drying rate (*U*) of the biomass is calculated using the following equation [27]:

$$U = \frac{dMR}{dt} \tag{4}$$

where *t* is the drying time.

The air humidity at outlet ($Y_{out}$) is calculated using the following equation [13]:

$$Y_{out}(t) = \frac{6m_{s,dry}U}{AV_g\rho_p d_p} + Y_{in} \tag{5}$$

where $m_{s,dry}$ is the dry mass of the particle, kg; *A* is the surface area of the particle, m²; $V_g$ is the air mass flow rate, kg/s; $\rho_p$ is the particle density, kg/m³; $d_p$ is the particle diameter, m; $Y_{in}$ is air humidity of inlet air, g of water/kg of dry air.

To distinguish between gas-side and particle-side kinetics, the normalized drying rate is calculated using the following equation [13,29]:

$$v = \frac{U}{A\rho_g\beta(Y_{out,e}(T_p) - Y_{out}(t))} \tag{6}$$

where *v* represents the normalized drying rate; $\rho_g$ is the gas density, kg/m³; $\beta$ is the mass transfer coefficient, m/s; $Y_{out}(t)$ represents the outlet gas humidity at time *t*, g water/kg of dry air; $Y_{out,e}$ represents the outlet gas humidity under the hygroscopic equilibrium, g water/kg of dry air; $T_p$ is the temperature of the particle, °C.

In general, the measurement of several isotherms at different temperatures is advisable. At a given solid moisture content, the relative air humidity ratio ($\varphi$) is obtained and transformed into the equilibrium moisture content $Y_{out,e}$ by [15]:

$$Y_{out,e}(T_p) = \frac{W_w}{W_g} \cdot \frac{\varphi p_{sat}(T_p)}{P - \varphi p_{sat}(T_p)} \tag{7}$$

where $W_w$ is the molecular weight of water vapor, kg/kmol; $W_g$ is the molecular weight of dry air, kg/kmol; *P* is the total pressure, Pa; $P_{sat}$ is the saturation pressure, Pa.

According to the common definition, the normalized biomass moisture content is [17]:

$$\eta = \frac{M - M_e}{M_c - M_e} \tag{8}$$

where $M_c$ represents the moisture content of the sample at the critical point (end of the first drying period), kg of water/kg of dry biomass.

The drying process for biomass mostly occurs in the falling rate period. Fick's second law of diffusion, as shown in Equation (9), has been widely used to describe the drying process and interpret experimental drying data during the falling rate period as internal mass transfer controls the drying process [30]. The mathematical solution of Fick's second law for diffusion is shown in Equation (10) [31]:

$$\frac{\partial MR}{\partial t} = \nabla[\delta(\nabla MR)] \tag{9}$$

$$MR = \frac{8}{\pi^2} \sum_{n=0}^{\infty} \frac{1}{(2n+1)^2} \exp\left(-\frac{(2n+1)^2 \pi^2 \delta t}{4L^2}\right) \tag{10}$$

where $n$ is a positive integer, $t$ is the drying time, s; $L$ is the half thickness of the sample, m.

When sample shrinkage is negligible, the initial moisture content distribution is uniform and constant moisture diffusivity is assumed; Equation (10) is suitable for determining $\delta$. This equation can be further simplified into Equation (11) by taking the first term of a series solution as follows [31,32]:

$$\ln(MR) = \ln(\frac{8}{\pi^2}) - \left(\frac{\pi^2 \delta}{4L^2}t\right) \tag{11}$$

The mass transfer coefficient ($\beta$) between the air and the particles can be determined from the following relationships for laminar flow and turbulent flow, respectively [13,16]:

$$\beta = \frac{Sh\delta}{d_p} = \frac{0.332 Re^{0.5} Sc^{0.33} \delta}{d_p} \tag{12}$$

and

$$\beta = \frac{Sh\delta}{d_p} = \frac{0296 Re^{4/5} Sc^{1/3} \delta}{d_p} \tag{13}$$

$$Re = \frac{u_g d_p}{\nu_g} \tag{14}$$

$$Sc = \frac{\nu_g}{\delta_g} \tag{15}$$

where $Sh$ is the Sherwood number; $Sc$ is the Schmidt number; $Re$ is the Reynold number; $\nu_g$ is the coefficient of kinematic viscosity for dry air, m²/s; $u_g$ is the gas velocity, m/s.

## 3. Results and Discussion

### 3.1. Effect of Drying Temperature

The effects of the drying temperature on the drying curves of the biomass are shown in Figure 2. The drying temperature significantly affected the moisture change in the biomass (Figure 2a). By increasing the air temperature, the biomass moisture content became steeper and the residual solid moisture content also became lower. The initial moisture content of the biomass was determined as 0.28 on a dry basis. The drying temperature was the main driving force of drying. Therefore, the high drying temperature could remove more moisture during drying; for instance, the final moisture content was 0.0085 for drying at 70 °C, whereas it was only 0.0005 for drying at 130 °C, a decline of 94.11%. Similar results have been reported by other authors [13,16].

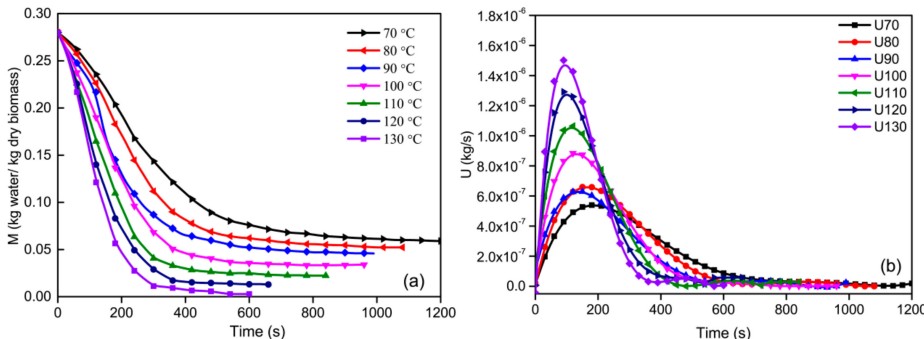

**Figure 2.** Effects of dying temperature on biomass drying process: (**a**) drying curves; (**b**) drying rate curves.

As shown in Figure 2b, there was no constant rate period in any drying conditions; this phenomenon probably resulted from the fact that the thickness of the biomass was unable to provide a constant supply of moisture. Similar results have been reported by different authors [33]. Therefore, the drying process could be divided into two periods, namely, the rising rate period (0~180 s) and the falling rate period (180~1200 s). The rising rate period was very short and only occurred at the beginning of the drying process. This short period of rising probably resulted from the increasing temperature of the biomass material, which directly improved the evaporation of free water on its surface. The falling rate period was the main drying process, during which internal diffusion dominated the moisture transfer in the material [26]. Similar results have been reported by other authors [26,31,34]. In addition, the drying rate significantly increased with the increase in the drying temperature; the drying rate was $5.34 \times 10^{-7}$ kg/s at 70 °C, whereas it was only $1.50 \times 10^{-6}$ kg/s at 130 °C, an approximate increase of three times. The results of the experiment further confirmed that the drying temperature was the main driving force of drying.

### 3.2. Effect of Initial Moisture Content

Figure 3 shows the effects of initial moisture content on the biomass drying process at 120 °C. As shown in Figure 3a, the moisture content at the end of drying increased with the increase in $M_o$ in the tested range. For example, the final moisture content was 0.01736 for drying at $M_o = 0.45$, whereas it was only 0.00741 for $M_o = 0.20$, as decrease of 57.32%. This phenomenon probably resulted from the fact that the drying time increased with the increase in $M_o$. Therefore, the final moisture content increased with the increase in $M_o$ within the same time. Similar results have been reported by other authors [19].

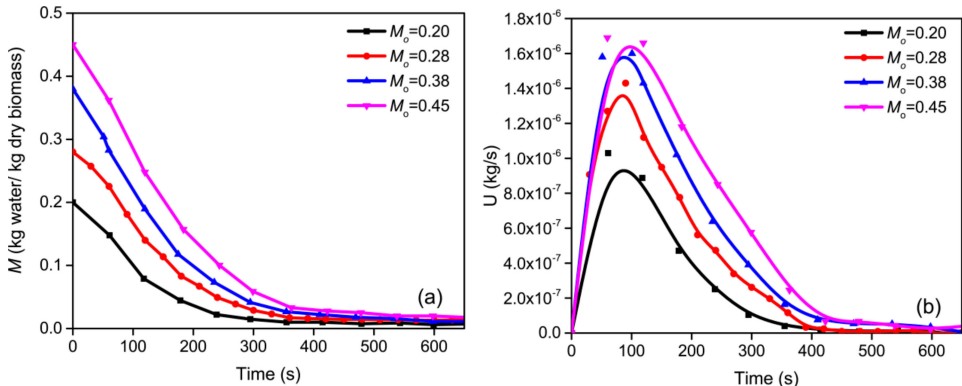

**Figure 3.** Effects of initial moisture content on biomass drying process: (**a**) drying curves; (**b**) drying rate curves.

As shown in Figure 3b, similar to Figure 2b, the evolution of the drying process for different initial moisture contents could also be divided into two periods. As the initial moisture content increased, the drying process was extended. In addition, the drying rate significantly increased with the increase in $M_o$; the drying rate was $4.17 \times 10^{-6}$ kg/s for $M_o = 0.45$, whereas it was only $1.25 \times 10^{-6}$ kg/s for $M_o = 0.20$, a decrease of 70.02%. Compared to the drying temperature (increased three times), the drying temperature had a more remarkable influence on the drying rate. At the same time, the dry time at the maximal value of the drying rate also increased. These results indicated that the content of free water for the intraparticle increased with the increase in $M_o$; therefore, the pressure difference between the internal and external pressures of the particle increased at 120 °C, which directly improved the evaporation of free water on the biomass surface in the short rising rate period.

*3.3. Characteristics of Drying Curves*

3.3.1. Humidity of Outlet Air

　　Figure 4 shows the humidity of the outlet air during the drying process. The effects of the drying temperature on the humidity of the outlet air are illustrated in Figure 4a. As shown in Figure 4a, the evolutions of the humidity of the outlet air for the raw samples were similar to each other. There was no constant air humidity drying period in any of the drying conditions. The evolutions of the outlet air humidity could be divided into two periods, namely, the rising period and the falling period. As the drying temperature increased, the maximal humidity of the outlet air increased, and the rising period and the falling period were reduced. For example, the air humidity was 63.3 g of water/kg of dry air for 130 °C, whereas it was only 34.4 g of water/kg of dry air for 70 °C, a decrease of 45.66%. Compared to Figure 2b, the evolutions of the humidity of outlet air for the raw samples were similar to the evolutions of the drying rates of the raw samples. This indicated that the humidity of the outlet air was obviously influenced by the drying rate. Figure 4b illustrates the effects of the initial moisture content on the humidity of the outlet air. As illustrated in Figure 4b, the humidity of the outlet air increased with the increase in $M_o$. However, the rising period and the falling period were extended. These results illustrated that the content of free water increased with the increase in $M_o$, the free water of the intraparticle more easily escaped outside, and the humidity of the outlet air increased.

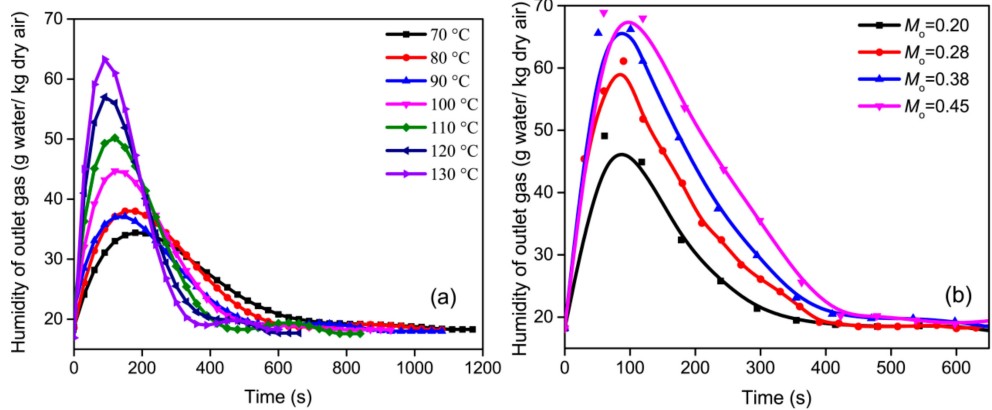

**Figure 4.** The humidity of outlet air during the drying process: (**a**) for different drying temperatures; (**b**) for different initial moisture contents.

3.3.2. Critical Moisture Content

　　The critical moisture content ($M_c$) is the dividing point between the rising rate period and the falling rate period, and is an important parameter in the design of a dryer [13]. The plots of the drying rate and the drying temperature for the biomass are shown in Figure 5a. As seen from Figure 5a, the drying process can be divided into three periods, namely, the rising rate period (0–0.15 kg water/kg dry air), the constant rate period (0.15–0.22 kg water/kg dry air), and the falling rate period (0.22~0.28 kg water/kg dry air). However, the constant drying rate period was not obvious in all drying conditions. The rising rate period was very short and only occurred at the beginning of the drying process, and the falling rate period was the main drying process. The results proved that internal diffusion dominated the moisture transfer in the biomass material, and the volume shrinkage of the biomass and the destruction of colloid caused the drying decrease during the falling rate period. Similar results have been reported by different authors [25,35]. The distribution of the critical moisture content for different drying temperatures is presented in Figure 5b. As illustrated from Figure 5b, as the drying temperature increased, the distribution of $M_c$ slowly increased first and then dramatically decreased. The maximal value of $M_c$ was 0.226 kg of water/kg of dry biomass for 80 °C, the minimal value of $M_c$ was 0.170 kg of water/kg of dry biomass for 130 °C, a decline of 24.78%. The distribution of $M_c$ for different

drying temperatures shows a non-linear relationship. The corresponding parabola fit, with $R^2 = 0.917$, was given by:

$$M_c = AT^2 + BT + C \tag{16}$$

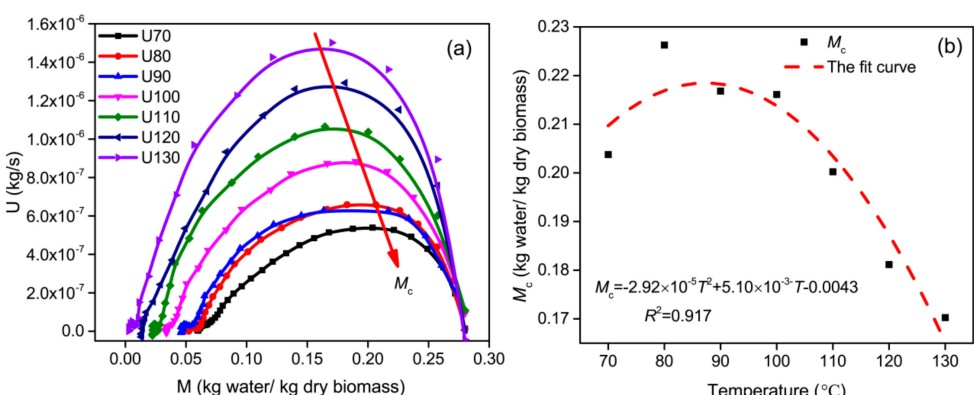

**Figure 5.** Distribution of critical moisture content for different drying temperatures: (**a**) the drying rate vs. the moisture content; (**b**) the critical moisture content vs. the drying temperature.

Figure 6a shows the drying rate curves of the biomass with different initial moisture contents. As illustrated in Figure 6a, increasing the initial moisture content in the tested range significantly increased the value of $M_c$. The distribution of critical moisture content for different $M_o$ is presented in Figure 6b. As seen from Figure 6b, the value of $M_c$ increased with the increase in $M_o$. The maximal value of $M_c$ was 0.362 kg of water/kg of dry biomass for $M_o = 0.45$, the minimal value of $M_c$ was 0.148 kg of water/kg of dry biomass for $M_o = 0.20$, a decline of 59.12%. Therefore, compared to the drying temperature (24.78%), the initial moisture content had a more remarkable impact on $M_c$. The parabolic relation between the critical moisture content and the initial moisture content was decided using the experimental fit method. The fit $R^2$-value of the fitting for $M_o$ ($R^2 = 0.997$) was higher than for the drying temperature ($R^2 = 0.917$). The corresponding parabola fit was given by:

$$M_c = AM_o^2 + BM_o + C \tag{17}$$

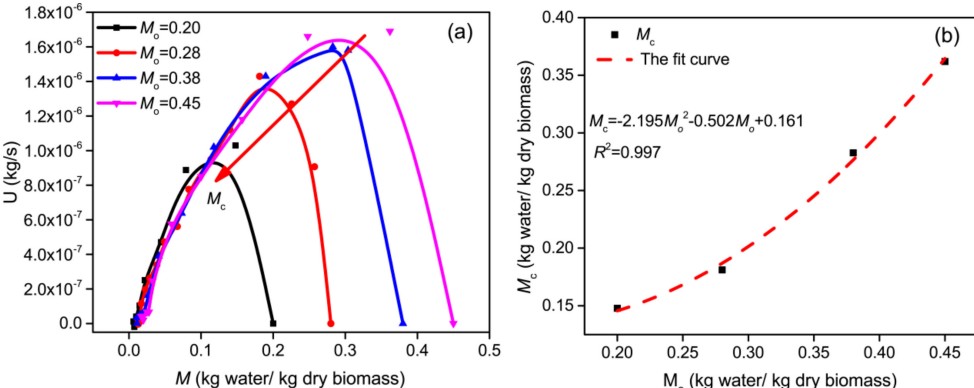

**Figure 6.** Distribution of critical moisture content for different initial moisture contents: (**a**) drying rate vs. moisture content; (**b**) critical moisture content vs. initial moisture content.

### 3.3.3. Effective Diffusivity

Figure 7 presents the distribution of effective diffusivity during the drying process. The distribution of effective diffusivity for different drying temperatures is illustrated in Figure 7a. From Figure 7a, it is obviously observed that the effective diffusivity of all of the tested samples constantly reduced with the increments in drying time. The evolutions of effective diffusivity could be divided into two periods, namely, the dramatically falling

period and the slowly falling period. The dramatically falling period was observed from 0 to 100 s in a narrow time range, in which the free water was rapidly evaporated. The drying temperature had a marginal effect on the effective diffusivity during the dramatically falling period. However, the effective diffusivity obviously increased with the increase in drying temperature during the slowly falling period. The plausible reason for this phenomenon could be explained by the fact that the internal temperature gradient was small, but the change in temperature was significant in a short time, so the internal thermal stresses were great enough to cause the particle to have a maximal value of $\delta$ at $t = 0$. As the drying time increased, the change in temperature decreased, and the internal thermal stresses declined. Therefore, the value of $\delta$ decreased. However, the internal thermal stresses were so great that the drying temperature had a marginal effect on the effective diffusivity during the dramatically falling period. The internal thermal stresses reached equilibrium with increased drying time. At this time, the internal thermal stresses had a marginal effect on the effective diffusivity during the slowly falling period. However, with the increase in gas temperature, the average kinetic energy of gas molecules increased, so the diffusion was accelerated. Therefore, the value of $\delta$ was increased during the slowly falling period.

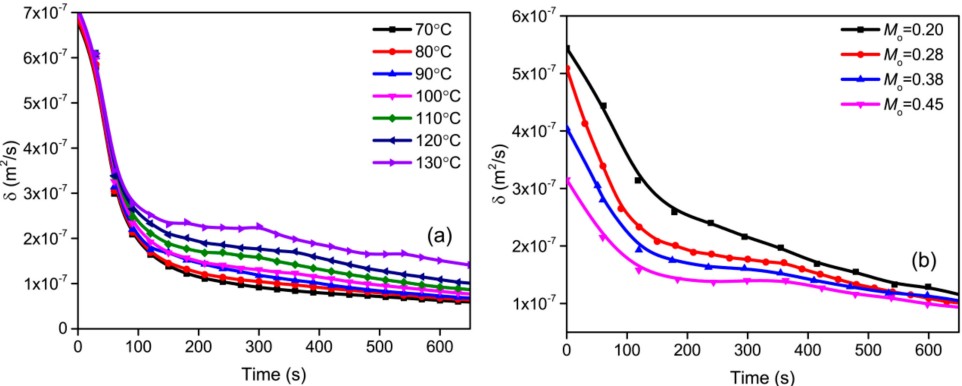

**Figure 7.** Evolutions of the distribution of effective diffusivity ($\delta$) with drying time for: (**a**) drying temperature; (**b**) initial moisture content.

The distribution of effective diffusivity for various $M_o$ is presented in Figure 7b. As shown in Figure 7b, the evolutions of effective diffusivity for various $M_o$ were similar to the evolutions of effective diffusivity for various drying temperatures, and could also be divided into two periods (the dramatically falling period and the slowly falling period). The initial moisture content had a significant influence on effective diffusivity during the dramatically falling period, and had a marginal effect on the effective diffusivity during the slowly falling period, which was contrary to the results for the drying temperature, as illustrated in Figure 7a. The plausible reason for this phenomenon can be explained by the fact that the interior particle was similar to a confined space, and the pressure of the interior particle rapidly increased with the free water evaporation during the dramatically falling period. However, when the pressure of the interior particle increased, the average free distance between molecules decreased, thus the effective diffusivity declined. Therefore, the value of $\delta$ decreased with the increase in $M_o$ during the dramatically falling period. The pressure gradient between the interior and the outside of the particle reached equilibrium with the increased drying time. At this time, the pressure of the interior particle had a marginal effect on the effective diffusivity during the slowly falling period. Therefore, the initial moisture content had a marginal effect on the effective diffusivity during the slowly falling period.

### 3.3.4. Coefficient of Mass Transfer

The coefficient of mass transfer can reflect the enhancement degree of the specific mass transfer process. Figure 8 shows the distribution of the mass transfer coefficient during the drying process. As presented in Figure 8, it was obviously observed that the mass

transfer coefficient of all of the tested samples was constantly reduced with the increments of drying time. These results illustrated that the effective diffusivity of the biomass material rapidly decreased with the increase in drying time. Therefore, the value of $\beta$ dramatically decreased. The evolutions of the mass transfer coefficient were similar to the evolutions of the effective diffusivity, as illustrated in Figure 8. The relationship between the mass transfer coefficient and the effective diffusivity is presented in Equation (12). The values of $Sh$ and $d_p$ were constant values with the increase in drying time; therefore, the value of $\beta$ had an obviously positive correlation with the value of $\delta$, and the evolutions of $\beta$ were similar to the evolutions of $\delta$.

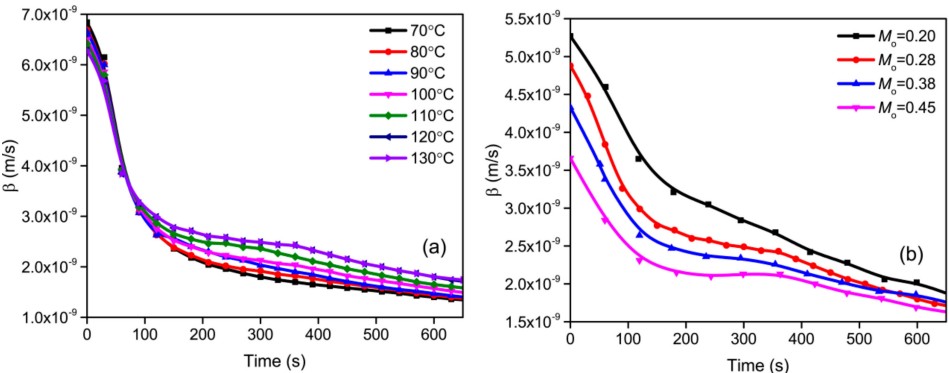

**Figure 8.** Distribution of mass transfer coefficient during the drying process: (**a**) for different drying temperatures; (**b**) for different initial moisture contents.

3.3.5. Normalized Drying Curves

The results of normalization are plotted in Figure 9. As Figure 9 shows, the drying temperature and initial moisture content had an obvious influence on the normalized single particle drying curves. Figure 9a presents the normalized single particle drying curves for various drying temperatures. From Figure 9a, the evolutions of the normalized single-particle drying curves could be divided into two periods, namely, the dramatically rising period ($\eta$ = 0~0.15) and the slowly rising period ($\eta$ = 0.15~1.00). When the normalized biomass moisture content reached 0.1, the influence on the normalized drying rate was the most noticeable. As the drying temperature increased, the value of $v$ generally increased. Specifically, the critical moisture was lower, and the value of $v$ was greater, as shown in Figure 5b. These results proved that there were remarkable negative relevant relations between the critical moisture and the normalized biomass moisture content. Figure 9b illustrates the results of normalization for various $M_o$. From Figure 9b, the evolutions of normalized single particle drying curves also had two periods (the dramatically rising period and the slowly rising period). Compared to Figure 9a, the dramatically rising period ($\eta$ = 0~0.05) became shorter. The results indicated that the initial moisture content had a marginal effect on the value of $v$ during the slowly rising period. The value of $v$ decreased with the increase in $M_o$. As shown in Figure 9b, the critical moisture decreased with the increase in $M_o$. This result further proved that the critical moisture and the normalized biomass moisture content obviously had a negative relevant relationship. Similar results have been reported by other authors [15,17].

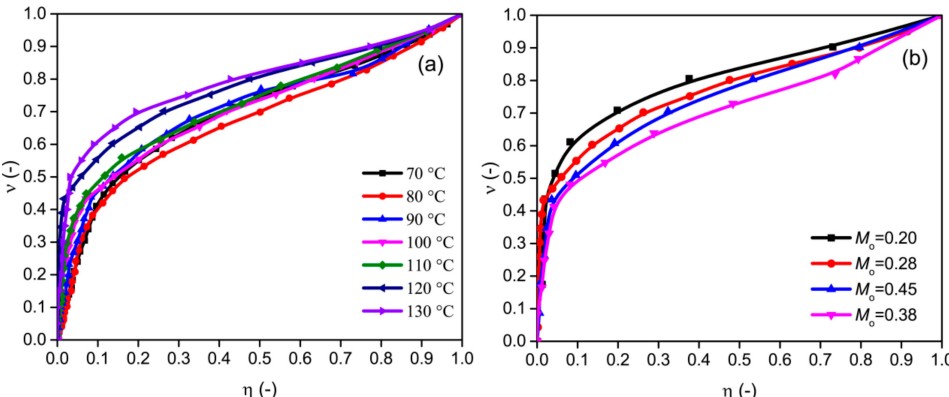

**Figure 9.** Normalized single particle drying curves: (**a**) for different drying temperatures; (**b**) for different initial moisture contents.

### 4. Conclusions

In this paper, Fick's second diffusion law was used to study the drying kinetics of a single biomass particle. The relationship between the critical moisture content and temperature was a parabolic relationship, which was similar to the relationship of the critical moisture content and the initial moisture content. The critical moisture content decreased with the increase in drying temperature and the initial moisture content. The maximal value of $M_c$ was 0.362 kg of water/kg of dry biomass for $M_o = 0.45$, the minimal value of $M_c$ was 0.148 kg of water/kg of dry biomass for $M_o = 0.20$, a decrease of 59.12%. The drying temperature had an effect on the effective diffusivity and the coefficient of mass transfer during the dramatically falling drying period; however, it was marginal during the slowly falling period. Compared to the drying temperature, the initial moisture showed the opposite effect for these different periods. The normalized biomass moisture content generally increased with the increase in drying temperature, and decreased with the increase in initial moisture content. The study concluded that the critical moisture content and the normalized biomass moisture content showed a negative relationship. This study will provide useful data which can act as an important reference for biomass drying.

**Author Contributions:** Conceptualization, J.C., D.H. and M.L.; methodology, L.Z., Q.W.; software, L.Z., Q.W.; formal analysis, J.C., L.Z., Q.W., X.T.; data curation, J.C., L.Z., Q.W., X.T.; writing—original draft preparation, J.C., L.Z., Q.W., X.T.; writing—review and editing, J.C., L.Z., Q.W., X.T.; funding acquisition, J.C., D.H. and M.L. All authors have read and agreed to the published version of the manuscript.

**Funding:** This research was funded by GuangXi Key Laboratory of New Energy and Building Energy Saving grant number (22-J-22-1 and 22-J-21-8); the Guangxi Natural Science Foundation of China (2020GXNSFBA297075 and AD20297010), the Guilin Scientific Research and Technology Development Plan of China (2021H0202 and 20210218-3), and the National Natural Science Foundation of China (52266011). And The APC was funded by GuangXi Key Laboratory of New Energy and Building Energy Saving grant number (22-J-22-1).

**Conflicts of Interest:** The authors declare no conflict of interest.

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
