# Peer review of "Drying Kinetics of a Single Biomass Particle Using Fick’s Second Law of Diffusion"

_processes, doi:10.3390/pr11040984_

Round 1
Reviewer 1 Report
Drying kinetics of single biomass particle using Fick’s second law of diffusion was studied in this paper. some minor issues should be addressed:
(1) The authors are suggested to use the jonurnal template.
(2) Page 2, "In this work, the drying kinetics of single biomass particle using Fick’s second law of diffusion was investigated by a hot air circulating oven." What I concern is that, If there are some other diffusion models that can describe the dehydration process of biomass particle?
(3)Page 7, "Fig. 4 presents the effects of initial moisture content and drying temperature on biomass drying process. As increased the initial moisture content", how the authors relized "increased the initial moisture content"? by soaking the biomass particle in water directly ?
(4) Page 7, Figure 4 seems puzzled. The ordinate is the drying time in seconds, what's the meaning for different temperature and initial moisture content?
Author Response
|
Reviewer comments |
Author' response |
Revised texts |
|
(1) The authors are suggested to use the journal template. |
Revised |
Thanks a lot for your kind proposal.
According to the Reviewer’s comment, we have revised the authors template based on the Processes template. |
|
|
|
|
|
(2) Page 2, "In this work, the drying kinetics of single biomass particle using Fick’s second law of diffusion was investigated by a hot air circulating oven." What I concern is that, If there are some other diffusion models that can describe the dehydration process of biomass particle? |
Revised |
Thank you for providing these insights.
This is an interesting perspective. According to your advice, we added the related content in the revised manuscript. “The mathematical models of the drying process are generally divided into the distributed model and the lumped parameter model [1]. The distributed model considers both internal and external heat and mass transport and can predict temperature and moisture gradients in the material. These models rely on Luikov equations [2], and the equations are derived from the Fick’s second law [3]. It use the irreversible transport law of thermodynamics, and start from the basic relationship of mass, momentum and energy conservation, assuming that temperature gradient and concentration gradient work together on water diffusion. Finally, a set of partial differential equations are defined to describe the relationship of the internal heat and mass transfer in materials. In most drying processes, the influence of pressure is much smaller than the influence of temperature and moisture content on the drying process of materials, so the pressure equation is generally ignored. The lumped parameter model does not focus on the temperature gradient in the material. It assumes a uniform temperature distribution and that the material temperature is equal to the drying medium temperature. In the lumped parameter model, the uniform temperature distribution and the sample temperature coincident with the ambient temperature, these two assumptions lead to some calculation errors [4]. These errors will be obvious at the beginning of drying, and reducing the thickness of the material will significantly reduce the calculation error. Therefore, this manuscript used Fick’s second law to build the mathematical models of the drying process.”. Introduction (Page 2).
|
|
|
|
|
|
(3) Page 7, "Fig. 4 presents the effects of initial moisture content and drying temperature on biomass drying process. As increased the initial moisture content", how the authors relized "increased the initial moisture content"? by soaking the biomass particle in water directly? |
Revised. |
Thanks a lot for your kind proposal.
This is an interesting perspective. The moisture content was decided by soaking the dried biomass particle in water directly. And then the initial moisture content of biomass was determined by means of AOAC method No. 934.06 (AOAC, 1990).
“The pre -processing moisture content was decided by soaking the dried biomass particle in water directly. The final initial moisture content of biomass was determined by means of AOAC method No. 934.06 (AOAC, 1990).” was added in the revised manuscript.
2.1. Materials
|
|
|
|
|
|
(4) Page 7, Figure 4 seems puzzled. The ordinate is the drying time in seconds, what's the meaning for different temperature and initial moisture content? |
Revised |
Thanks a lot for your kind proposal.
We are sorry to cause such troubles. We have deleted the related content in the revised manuscript.
|
|
|
|
|

Reviewer 2 Report
1- I propose the authors to use up-to-date references published in recent years.
2- In the conclusion section, more explanations are necessary to illustrate the application of useful data (for designing and operating of dryers for biomass) presented in the manuscript.
3- I propose the authors to Illustrate the curves for presenting the correlation of the experimental data with the theoretical data based on the equations represented in the manuscript.
4- Authors should introduce the software or mathematical methods used to extract the coefficient of mass transfer and effective diffusivity.
5- The novelty of research should be illustrated in more details in the introduction section.
Author Response
|
Reviewer comments |
Author' response |
Revised texts |
|
(1) I propose the authors to use up-to-date references published in recent years. |
Revised |
Thanks a lot for your kind proposal.
We regret there were problems with the references. Some references in the manuscript has been updated references published in recent years.
|
|
|
|
|
|
(2) In the conclusion section, more explanations are necessary to illustrate the application of useful data (for designing and operating of dryers for biomass) presented in the manuscript. |
Revised |
Thanks a lot for your kind proposal.
According to your comments, we have revised the manuscript.
The conclusion was revised to “In this paper, the Fick's second diffusion law was used to study the drying kinetics of a single biomass particle. The relation between critical moisture content and temperature was parabolic relationship, which was similar to the relationship of critical moisture content and initial moisture content. The critical moisture content decreased with the increasing of drying temperature and the initial moisture content. The maximal value of Mc was 0.362 kg water/kg dry biomass for Mo=0.45, the minimal value of Mc was 0.148 kg water/kg dry biomass for Mo=0.20, declined by 59.12%. The drying temperature had effects on the effective diffusivity and coefficient of mass transfer during the dramatically falling drying period; however, it was marginal during the slowly falling period. Compared to drying temperature, the initial moisture showed the opposite effect for those different periods. The normalized biomass moisture content generally increased with the increasing of drying temperature, and decreased with the increasing of initial moisture content. The study concluded that the critical moisture content and the normalized biomass moisture content showed negative relation. This study will provide to gather useful data which can provide important reference for biomass drying.”
4. Conclusion.
|
|
|
|
|
|
(3) I propose the authors to Illustrate the curves for presenting the correlation of the experimental data with the theoretical data based on the equations represented in the manuscript. |
Responded |
Thank you for providing these insights.
This is an interesting perspective. This study calculated the coefficient of mass transfer of biomass single particles under different temperature and initial moisture content through experimental data and the second law of Fick. Because the mathematical model was not revised in the calculation process, we would revise the model in the future work. |
|
|
|
|
|
(4) Authors should introduce the software or mathematical methods used to extract the coefficient of mass transfer and effective diffusivity. |
Revised |
Thank you for providing these insights.
According to your comments, we have added we added the related content in the revised manuscript. “The mathematical models of the drying process are generally divided into the distributed model and the lumped parameter model [1]. The distributed model considers both internal and external heat and mass transport and can predict temperature and moisture gradients in the material. These models rely on Luikov equations [2], and the equations are derived from the Fick’s second law [3]. It use the irreversible transport law of thermodynamics, and start from the basic relationship of mass, momentum and energy conservation, assuming that temperature gradient and concentration gradient work together on water diffusion. Finally, a set of partial differential equations are defined to describe the relationship of the internal heat and mass transfer in materials. In most drying processes, the influence of pressure is much smaller than the influence of temperature and moisture content on the drying process of materials, so the pressure equation is generally ignored. The lumped parameter model does not focus on the temperature gradient in the material. It assumes a uniform temperature distribution and that the material temperature is equal to the drying medium temperature. In the lumped parameter model, the uniform temperature distribution and the sample temperature coincident with the ambient temperature, these two assumptions lead to some calculation errors [4]. These errors will be obvious at the beginning of drying, and reducing the thickness of the material will significantly reduce the calculation error. Therefore, this manuscript used Fick’s second law to build the mathematical models of the drying process.”. Introduction (Page 2).
|
|
|
|
|
|
(5) The novelty of research should be illustrated in more details in the introduction section. |
Revised |
Thank you for providing these insights.
This is an interesting perspective. According to your advice, we added the related content in the revised manuscript. “The mathematical models of the drying process are generally divided into the distributed model and the lumped parameter model [1]. The distributed model considers both internal and external heat and mass transport and can predict temperature and moisture gradients in the material. These models rely on Luikov equations [2], and the equations are derived from the Fick’s second law [3]. It use the irreversible transport law of thermodynamics, and start from the basic relationship of mass, momentum and energy conservation, assuming that temperature gradient and concentration gradient work together on water diffusion. Finally, a set of partial differential equations are defined to describe the relationship of the internal heat and mass transfer in materials. In most drying processes, the influence of pressure is much smaller than the influence of temperature and moisture content on the drying process of materials, so the pressure equation is generally ignored. The lumped parameter model does not focus on the temperature gradient in the material. It assumes a uniform temperature distribution and that the material temperature is equal to the drying medium temperature. In the lumped parameter model, the uniform temperature distribution and the sample temperature coincident with the ambient temperature, these two assumptions lead to some calculation errors [4]. These errors will be obvious at the beginning of drying, and reducing the thickness of the material will significantly reduce the calculation error. Therefore, this manuscript used Fick’s second law to build the mathematical models of the drying process.”. Introduction (Page 2).
|
|
|
|
|
